# Emotional Eating, Health Behaviours, and Obesity in Children: A 12-Country Cross-Sectional Study

**DOI:** 10.3390/nu11020351

**Published:** 2019-02-07

**Authors:** Elli Jalo, Hanna Konttinen, Henna Vepsäläinen, Jean-Philippe Chaput, Gang Hu, Carol Maher, José Maia, Olga L. Sarmiento, Martyn Standage, Catrine Tudor-Locke, Peter T. Katzmarzyk, Mikael Fogelholm

**Affiliations:** 1Department of Food and Nutrition, University of Helsinki, 00014 Helsinki, Finland; hanna.konttinen@helsinki.fi (H.K.); henna.vepsalainen@helsinki.fi (H.V.); mikael.fogelholm@helsinki.fi (M.F.); 2Sociology, University of Helsinki, 00014 Helsinki, Finland; 3Children’s Hospital of Eastern Ontario Research Institute, Ottawa, ON K1H 8L1, Canada; jpchaput@cheo.on.ca; 4Pennington Biomedical Research Center, Baton Rouge, LA 70808, USA; gang.hu@pbrc.edu (G.H.); peter.katzmarzyk@pbrc.edu (P.T.K.); 5Alliance for Research In Exercise Nutrition and Activity (ARENA), School of Health Sciences, University of South Australia, Adelaide, SA 5001, Australia; carol.maher@unisa.edu.au; 6CIFI2D, Faculdade de Desporto, University of Porto, 4200-450 Porto, Portugal; jmaia@fade.up.pt; 7School of Medicine, Universidad de los Andes, Bogotá 11001000, Colombia; osarmien@uniandes.edu.co; 8Department for Health, University of Bath, Bath BA2 7AY, UK; m.standage@bath.ac.uk; 9Department of Kinesiology, School of Public Health and Health Sciences, University of Massachusetts Amherst, MA 01003, USA; ctudorlocke@umass.edu

**Keywords:** eating behaviour, psychological eating style, negative emotions, Emotion-Induced Eating Scale, health behaviour, BMI

## Abstract

Eating in response to negative emotions (emotional eating, EE) may predispose an individual to obesity. Yet, it is not well known how EE in children is associated with body mass index (BMI) and health behaviours (i.e., diet, physical activity, sleep, and TV-viewing). In the present study, we examined these associations in a cross-sectional sample of 5426 (54% girls) 9–11-year-old children from 12 countries and five continents. EE, food consumption, and TV-viewing were measured using self-administered questionnaires, and physical activity and nocturnal sleep duration were measured with accelerometers. BMI was calculated using measured weights and heights. EE factor scores were computed using confirmatory factor analysis, and dietary patterns were identified using principal components analysis. The associations of EE with health behaviours and BMI *z*-scores were analyzed using multilevel models including age, gender, and household income as covariates. EE was positively and consistently (across 12 study sites) associated with an unhealthy dietary pattern (β = 0.29, SE = 0.02, *p* < 0.0001), suggesting that the association is not restricted to Western countries. Positive associations between EE and physical activity and TV viewing were not consistent across sites. Results tended to be similar in boys and girls. EE was unrelated to BMI in this sample, but prospective studies are needed to determine whether higher EE in children predicts the development of undesirable dietary patterns and obesity over time.

## 1. Introduction

Childhood obesity rates are high in both developed and developing countries [1]. It is likely that the most important contributors are the increased availability of energy-dense foods and a reduced need for physical activity—the current obesogenic environment. Many individual characteristics could be relevant for explaining the differential susceptibility to the development of obesity among individuals in the same environment. One example is emotional eating (EE), which refers to a tendency to eat more in response to negative emotions [2,3]. According to the EE theory (also called the psychosomatic theory), individuals with EE use eating to reduce the intensity of negative emotions [2]. This is considered a poor coping strategy, and such difficulties in emotion regulation may be one possible mechanism underlying EE [4]. Foods consumed in response to negative emotions are usually high in sugar and/or fat [3]. These palatable foods provide hedonic pleasure and instant reward, which may distract from the experience of negative emotions [3]. Because an expected normal physiological reaction to negative emotions is a suppressed appetite [3,5], EE may interfere with physiological regulation. It may therefore represent a risk factor for becoming overweight and obese. Several studies have indeed suggested this might be the case in adults, since EE has been found to correlate positively with body mass index (BMI) [6,7,8,9] and to predict weight gain [10,11]. 

However, in children, empirical evidence regarding the association between EE and obesity is far from conclusive. Cross-sectional studies conducted with children (mean age between 7 and 13 years) have reported a positive association [12,13,14,15,16,17,18], no association [19,20], or even an inverse association [21,22,23] between EE and BMI/being overweight. It is possible that some of these inconsistencies are related to the use of different approaches to measure EE in previous work. Even though it has been shown that there is a good agreement between self-reported and parent-reported EE [13], the majority of the studies reporting positive associations with BMI/being overweight have employed parent-reported EE [12,14,15,16,17,18]. In contrast, self-reported EE has been employed in studies reporting an inverse association with BMI/being overweight [21,22,23]. Regardless, broader measures of emotion dysregulation have also been associated with obesity in children. For example, emotion-driven impulsiveness was associated with increased BMI *z*-scores in a large sample of 12–18-year-old children [24]. In longitudinal studies, parent-reported EE at the age of 5 to 6 years predicted higher BMI in 7–8-year-old children [25], but parent-reported EE at the age of 6 years was not associated with changes in BMI standard deviation scores in children aged 6 to 8 years [26].

Health behaviours, such as adhering to a healthy diet and getting adequate physical activity and sleep, are potentially important in prevention of childhood obesity [27,28,29]. As in adults, EE has been associated with a higher consumption of salty and sweet energy-dense foods and soft drinks in 12–15-year-old children [30] and a higher consumption of sweets and soft drinks in 12-year-old girls [31]. In contrast, in children aged between 5 and 12 years, no association between EE and the consumption of snacks [20,32], sweet foods [32], or fatty foods [32] has been reported. Even though these contradictory findings may be due to methodological issues (e.g., crude measure for snacking, and parent- vs. self-reported food consumption data), it is currently unclear whether EE is associated with diet in children under the age of 12 years. The clustering of health behaviours in children [33,34] raises the question of whether EE is also related to physical activity, sedentary behaviour (such as TV viewing), and/or sleep duration. Given that these behaviours do not occur in isolation, it is important to study them simultaneously. 

The aim of this study was to examine the associations between self-reported EE, health behaviours (i.e., dietary patterns, physical activity, sleep duration, and TV viewing), and BMI in 9–11-year-old children. Because of inconsistent evidence and the limited number of previous studies, the analyses were exploratory, and we did not have specific hypotheses regarding the directions of these associations. Previous studies on EE in children have been mainly conducted in Western countries (in Europe and North America). In this work, we used a large sample from 12 countries and five continents, which gave us a unique opportunity to examine whether EE is consistently linked to behaviours that predispose individuals to obesity across countries representing diversity in terms of development, culture, socioeconomic status, and ethnic backgrounds.

## 2. Materials and Methods 

### 2.1. Study Setting and Participants

The present study is a secondary analysis of the International Study of Childhood Obesity, Lifestyle and the Environment (ISCOLE), which aimed to determine the relationships between lifestyle behaviours and obesity in children. The details of the ISCOLE protocol have been reported previously [35]. The cross-sectional sample consisted of 9–11-year-old children from study sites located in urban and semi-urban areas in 12 different countries from all parts of the world (Australia, Brazil, Canada, China, Colombia, Finland, India, Kenya, Portugal, South Africa, the United Kingdom, and the United States). Rural areas were excluded due to logistical concerns. Each study site identified one or more school districts with a sufficient population to provide a sample of 500 children. The primary sampling unit within sites was schools, and the secondary sampling unit was classes within the schools. Schools were stratified by socio-economic status before sampling in order to maximize variability within sites see [35]. The Institutional Review Board at the Pennington Biomedical Research Center (coordinating center) approved the overarching ISCOLE protocol, and the Institutional/Ethical Review Boards at each participating institution approved the local protocols. Parents or legal guardians provided written informed consent, and children provided their written assent before participation, as required by local ethics boards. Data were collected from September 2011 through December 2013.

In total, 7372 children participated in ISCOLE [36], of which 5426 (74%) were included in the present analytical sample. Children who were missing data on one or several of the following variables were excluded: emotional eating (0.7% were excluded due to lacking data on this variable), dietary patterns (2.3%), accelerometry (moderate to vigorous physical activity (MVPA) and/or sleep, 16.5%), TV viewing (0.6%), BMI (0.4%), and household income (11.1%).

### 2.2. Emotional Eating

EE was measured via the Emotion-Induced Eating Scale (EIES), which was developed using a sample of over 2000 girls aged 9 to 10 years from the United States by Striegel-Moore et al. [21]. Children completed the EIES as part of a six-page questionnaire. The original EIES consists of seven items pertaining to emotionally-induced eating, as follows: eating in response to feeling sad, worried, mad, bored, or happy (e.g., “I eat more when I’m sad”), eating when not hungry, and using food as a reward. Responses were recorded on a 3-point scale (1 = never or almost never, 2 = sometimes, 3 = usually or always). See Appendix A for item level frequencies of the original EIES. However, taking a closer look at the wordings of the original EIES items reveals that two items out of the seven do not describe eating in response to emotions (“I eat between meals even if I’m not hungry” and “When I do something well I give myself a food treat”), and one item describes eating in response to a positive affective state (“I eat more when I’m happy”). EE is traditionally defined as “eating induced by negative emotions” [2], and it is also suggested that the desire to eat in response to negative emotions vs. positive emotions are two different constructs [5]. We wanted to adhere to this definition, and therefore used only the four items describing eating in response to negative emotions (sad, mad, worried, and bored) to assess EE in this study.

A confirmatory factor analysis (CFA) using a weighted least squares estimation with robust standard errors and a mean and variance adjusted test statistic (Lavaan package version 0.5.-23.1097 in the R system of statistical computing) was applied to test whether the four items describing eating in response to negative emotions loaded onto a one-factor latent construct. The items were used as ordinal variables in the analysis. The CFA was first conducted for the pooled data and then repeated for each study site and both genders separately. When evaluating the model fit, Comparative Fit Index (CFI) and Tucker–Lewis Index (TLI) values ≥0.95, a Root Mean Square Error of the Approximation (RMSEA) value ≤0.06, and a Standardized Root Square Mean Residuals (SRMR) value ≤0.08 were considered to indicate a good fit between the model and data [37]. Based on these criteria, the model showed an excellent fit for the pooled data (χ^2^(df) = 2.660(2), *p* for χ^2^ = 0.264, CFI = 1.000, TLI = 1.000, RMSEA = 0.007 and SRMR = 0.006, Table 1). The Cronbach’s alpha value for the four EE items was 0.60. For comparison, we tested also the original 7-item scale. CFA indicated a poorer fit compared to 4-item model (χ^2^(df) = 364.68(14), *p* for χ^2^ < 0.001, CFI = 0.96, TLI = 0.94, RMSEA = 0.06, and SRMR = 0.04). Furthermore, the Cronbach’s alpha value for the 7-item scale (0.65) was not substantially higher than the respective value for the 4-item scale, especially because alpha always increases when more items are added [38]. These results further supported our initial decision to include only the four items measuring eating in response to negative emotions.

In the gender-stratified analysis, standardized factor loadings were highly comparable between boys and girls (Table 1). The test of the measurement invariance also indicated strong invariance, since CFI values changed less than 0.01 [39] when loadings (ΔCFI = 0.001, *p* for Δχ^2^ = 0.416), thresholds (ΔCFI < 0.001, *p* for Δχ^2^ = 0.874), and residual variances (ΔCFI = 0.001, *p* for Δχ^2^ = 0.182) were forced equal across the genders in a stepwise manner. There were some differences in factor loadings between the study sites in the site-stratified analysis, especially regarding the item about eating in response to boredom (factor loadings ranged from 0.31 to 0.87 between the sites). The test of the measurement invariance also indicated that loadings were not equal across the study sites (ΔCFI = 0.03, *p* for Δχ^2^ < 0.001). Yet, since partial metric invariance (equality of factor structure and factor loadings) was achieved after releasing an equality constraint only for one factor loading (the “bored” item, ΔCFI=0.008, *p* for Δχ^2^ = 0.064), we decided to use the EE factor scores from the pooled data in the present analyses. However, we also repeated all analyses by using the site-specific EE factor scores and found that the results remained highly similar (data not shown). EE factor scores were computed using the Empirical Bayes approach.

### 2.3. Dietary patterns

Dietary patterns were defined using data from a self-administered food frequency questionnaire (FFQ), in which children reported their usual consumption frequency of 23 different food groups, according to seven response categories ranging from “never” to “more than once a day”. The FFQ was validated within ISCOLE against 3-day pre-coded food diaries [40]. The identification of the two dietary patterns has been reported in detail elsewhere [41]. In short, principal components analysis (PCA) with an orthogonal Varimax transformation was carried out using weekly portions of the FFQ food groups as input variables. From the 23 FFQ food groups, fruit juices were excluded from PCA due to low validity of reporting [40]. Two components were identified and named: (1) an unhealthy diet pattern, with high loadings for fast foods, ice cream, fried food, French fries, potato chips, cakes and sugar-sweetened sodas, and (2) a healthy diet pattern, with high loadings for dark-green vegetables, orange vegetables, vegetables in general, and fruits and berries. The naming was based on previous knowledge of associations between health and food items that loaded highly on the two dietary patterns. In total, the two dietary patterns explained 36% of the total variance in reported food consumption. The unhealthy diet pattern was stronger with an eigenvalue of 4.8 (22% of variance explained), and the healthy diet pattern was slightly weaker with an eigenvalue of 3.1 (14% of variance explained). The PCA was also repeated for each site separately, and the resulting site-specific diet patterns were very similar to the patterns that emerged with the pooled data. In the present paper, we used the pattern scores from the pooled data in accordance with the EE factor scores.

### 2.4. Physical Activity and Sleep

The average daily time spent in MVPA and average nocturnal sleep duration were assessed using the Actigraph GT3X+ accelerometer (Pensacola, FL, USA). The device was worn at the waist on an elastic belt for 24 hours per day (removing only for water-related activities, such as swimming or taking a shower) for at least seven days. The minimal amount of data considered adequate to calculate the average daily MVPA was at least four days with 10 or more hours of daily awake wear time, including at least one weekend day. MVPA was defined as ≥574 counts per 15 s [42]. Nocturnal sleep duration was estimated using an algorithm for 24-h accelerometers that was previously validated for the ISCOLE [43]. This algorithm captures the total nocturnal sleep time from sleep onset to the end of sleep and distinguishes it from daytime sleep episodes, and its accuracy was shown to be acceptable when compared to sleep logs [43]. Only nights with valid sleep (total sleep time ≥160 min) were used to calculate the mean nocturnal sleep duration of the week and adequate data for calculating the average value was considered at least 3 valid nights, including one weekend night (Friday or Saturday). 

### 2.5. TV Viewing

TV viewing time was determined with a self-administered questionnaire, which was adapted from the U.S. Youth Risk Behaviour Surveillance system [44]. Children were asked how many hours they typically watched TV for weekdays and weekend days separately. The response categories (respective scores) were “I did not watch TV” (0), less than 1 h (0.5), 1 h (1), 2 h (2), 3 h (3), 4 h (4), and 5 or more h (5). The total score was calculated by weighing the responses for weekdays by 5/7 and weekend days by 2/7. The TV viewing questions were shown to have adequate reliability (*r* = 0.55–0.68) and validity (*r* = 0.47) in a sample of 11–15-year-old U.S. children [45].

### 2.6. BMI

Anthropometric measurements were conducted during the school day by trained study assistants. Height was measured without shoes using a Seca 213 portable stadiometer (Hamburg, Germany) and weight was measured when participants were barefoot, in light indoor clothing, and without any pocket items, using a portable Tanita SC-240 Body Composition Analyzer (Arlington Heights, IL). Two measurements were obtained, and the average was used for analysis. If the first two measurements were more than 0.5 cm or 0.5 kg apart for height and weight, respectively, a third measurement was done, and the closest two measurements were averaged for analysis. BMI was calculated dividing the weight by the height squared (kg/m^2^). Age- and gender-specific reference data from the World Health Organization were used to compute the BMI *z*-scores [46]. 

### 2.7. Covariates

Age, gender, and household income were used as covariates. Parents reported the annual household income using eight to ten predefined categories designed for each study site. Country-specific income categories were merged into four levels. It was not possible to achieve exact quartiles, but the aim of the merging was to ensure the distribution of income was as balanced as possible. 

### 2.8. Statistical analyses

The descriptive results were calculated using IBM SPSS Statistics 24 (IBM SPSS, Chicago, IL, USA). An independent samples t-test and a χ^2^ test were used to compare the analytical sample with excluded children. Since the sample was clustered at three levels (students nested within schools nested within study sites), the associations between the EE factor scores and dependent variables were assessed using multilevel linear regression models (PROC MIXED of SAS statistical package version 9.4; SAS Institute Inc., Cary; NC, USA). Study sites and schools nested within study sites were both considered to have random effects. The denominator degrees of freedom for statistical tests pertaining to fixed effects were calculated using the Kenward and Roger approximation [47]. The first adjusted models included age, gender, and household income as covariates. In addition to that, second fully adjusted models included other health behaviours and BMI *z*-scores. Since a few earlier studies found a gender difference in the association between EE and BMI [22,23], we tested the interaction between the EE factor scores and gender in the fully adjusted models. Multilevel linear regression models were repeated for each site separately, and in these analyses, schools were considered as having random effects. Due to the skewness of the EE factor score variable, it was also studied as a categorical variable with five categories. This variable was formed with two steps: first, children who answered “never” for all four EE items were categorized into one group, and second, remaining children were categorized into quarters based on the EE factor scores.

## 3. Results

### 3.1. Descriptive Results

The analytical sample of the present study comprised 5426 children (74% of the overall study sample). Children who were excluded because of missing data showed more tendency towards EE than the included children (mean (SD) score 0.09 (0.58) vs. 0.03 (0.56), *p* < 0.001). In addition, as compared with the analytical sample, they had higher scores for the unhealthy diet pattern (0.22 (1.18) vs. −0.07 (0.93), *p* < 0.001), watched more TV (TV viewing score 1.8 (1.3) vs. 1.7 (1.2), *p* = 0.001), and had higher BMI *z* scores (0.61 (1.28) vs. 0.44 (1.25), *p* < 0.001). Also, more boys were excluded from the analytical sample than girls (28% vs. 25%, *p* = 0.002). Parents of the excluded children belonged more frequently to the lowest income group compared with those in the analytical sample (30% vs. 25%, *p* < 0.001). There were no differences in the scores for the healthy diet pattern, the amount of MVPA, sleep duration, or age between the excluded and included children.

Out of the analytical sample, 32% of the children answered “never or almost never” to all four EE items, and 25% answered “never or almost never” to three items and “sometimes” to one item. Only 18 (0.3%) children answered “usually or always” to all four EE items. Descriptive results stratified by gender and study site including the EE factor scores are presented in Table 2. There were no significant differences in the EE factor scores between boys and girls (data not shown), except in the United Kingdom, where girls showed a higher tendency towards EE compared with boys (mean (SD) score 0.10 (0.55) in girls vs. −0.04 (0.50) in boys, *t*-test = −2.37, *p* = 0.018). 

### 3.2. Associations between Emotional Eating, Health Behaviours and BMI 

The results from the multilevel linear regression models using the EE score as an independent variable are presented in Table 3. In the unadjusted models, EE was positively associated with the unhealthy diet pattern, MVPA, and TV viewing, and inversely associated with the BMI *z*-score. No associations were observed between the EE and the healthy diet pattern or sleep duration. The associations with the unhealthy diet pattern, MVPA, and TV viewing remained significant after adjusting for covariates (age, gender, and income), other health behaviours, and BMI *z*-scores. There was an interaction between the EE score and gender only in association with MVPA (*p* = 0.02). The association between EE and MVPA was positive for both genders, but it was stronger in boys (β = 3.31, SE = 0.86, *p* = 0.0001) than in girls (β = 1.28, SE = 0.60, *p* = 0.035, Table 4). 

The adjusted multilevel linear regression models were repeated for each site separately (Table 4). The positive association observed between the EE score and the unhealthy diet pattern was consistent across the study sites, but the significant positive association between EE and MVPA was observed only in Colombia and South Africa, and the significant positive association between EE and TV viewing was observed only in Canada and China.

When EE was included in the adjusted multilevel linear regression models as a categorical variable to account for the skewness of the original continuous variable, the results remained highly similar (Table 5). Compared to children with no EE, children in all EE quarters had higher unhealthy diet pattern scores, and this association followed a linear pattern according to the EE score quarters. A linear positive association was also found between the categorical EE variable and MVPA, and there was a significant, but not very strong, interaction with gender (*p* = 0.04). The association was significant for both genders, but it was stronger for boys. The categorical EE variable was also associated with TV viewing, but the association was not linear. Similar to the continuous EE score, the categorical EE variable was not associated with the healthy diet pattern, sleep duration, or BMI *z*-score.

## 4. Discussion

In this large, international sample of 9–11-year-old children, we found a positive association between EE and an unhealthy diet pattern. The novel finding of this study was that this positive association appeared to be consistent across the 12 study sites representing very different cultural and environmental settings. This is an important contribution to the current literature which has been almost completely focused on Western countries. We also found that EE was positively associated with both MVPA and TV viewing, yet these patterns were not consistent across all study sites. There were no associations between EE and healthy diet pattern, nocturnal sleep duration, or BMI *z*-score.

Our findings regarding the positive association between EE and the unhealthy diet pattern and the lack of association between EE and the healthy diet pattern support the earlier findings mainly in adults [3] that EE is especially associated with increased consumption of sweet and high-fat foods, which are generally considered to be highly palatable. Previous studies in 12–15-year-old children have also indicated that EE is associated with higher consumption of energy-dense foods and soft drinks, but it is not associated with the consumption of fruits and vegetables [30,31]. It must be noted that in addition to the palatability and nutrient composition of foods, food choice in response to emotional state might also reflect the availability and accessibility of foods. Many foods included in the unhealthy diet pattern (i.e., fast foods, ice cream, fried food, French fries, potato chips, cakes, and sugar-sweetened sodas) are ready to eat and are easily obtainable, even for 9–11-year-olds. However, one cause of EE may be poor emotion regulation in general [4], and it has been shown that adolescents (mean age 13.6 years) who are having difficulties with adequately regulating their negative emotions consume snacks, especially energy-dense snacks, more frequently [48]. Yet, because our study, as well as most of the earlier research regarding the association between emotional eating and diet, was cross-sectional, we cannot rule out the possibility of a reverse relationship. That might be the case if an unhealthier diet is associated with a decreased mood, or impaired hunger control, making it easier to eat in emotional states. 

Interestingly, in 5–12-year-old children, an age group closer to the present sample age range, no association between self-reported EE and food consumption has been reported previously [20,32]. One explanation for these contradictory findings may be differences in how diet was measured. Van Strien and Oosterveld [20] asked one simple question about the weekly frequency of consumption of sweet and/or savoury snacks with four answer options (never/sometimes/often/everyday), which might not have allowed enough variance to detect a possible association. In the study by Michels et al. [32], food consumption was reported by the parents, and it is possible that foods eaten in response to emotions are eaten without the knowledge of the parents. It has also been suggested that in children aged 6 to 18 years, self-reporting of dietary intake is more valid than parental reports [49,50]. 

As mentioned above, the positive association between EE and the unhealthy diet pattern appeared to be similar in all 12 countries. All beta estimates were positive, but there were some differences in the sizes of the estimates. The significance of these differences was not tested, because comparing each country against all of the remaining countries was outside of the scope of this study. Yet, similar findings across different countries suggest that the association between EE and the unhealthy diet pattern is not restricted to Western countries and their cultural and food environments. A normal physiological reaction to negative emotions is expected to be a suppressed appetite [5]. So, rather than being an innate characteristic, EE is most likely learned [51], and, for example, the home environment [52] and parenting [53] may affect this process. Our results suggest that the association between EE and unhealthy food consumption in children is independent, or at least not fully dependent, on the culture and culturally learned behavioural patterns. 

We found a small but significant positive association between EE and the amount of both daily MVPA and TV viewing. However, in contrast to the association with the unhealthy diet pattern, these associations were not consistent between all study sites. The significant positive associations between EE and these outcomes were observed only in two out of 12 sites (Colombia and South Africa for of MVPA and Canada and China for TV viewing). These findings need further clarification and replication in other study samples, because unassessed local factors may influence the observed relationships. For example, children undertake MVPA under a variety of contexts, and given that accelerometry was used to measure MVPA, we have detailed information on the level and pattern of MVPA but not the cultural, social or environmental context under which MVPA was performed. 

We hypothesized that EE might be related to MVPA and TV-viewing, because unhealthy behaviours tend to cluster in children [33,34]. The observed positive association between EE and MVPA was in contrast to that hypothesis. To our knowledge, only one study has previously examined the association between EE and physical activity in children, with no association between EE and weekly frequency of doing sports reported [20]. However, in that study, the amount of physical activity was measured with one simple question, whereas in the present study, MVPA was measured objectively using accelerometers. A few earlier studies support our finding regarding a positive association between EE and TV viewing. Ouwens et al. [54] reported a small but significant positive correlation between EE and TV viewing, and in another study, a significant positive correlation was found, but only in girls [20].

As the present study was cross-sectional, future studies are needed to determine the causal relationship and possible mechanisms for the association between EE and both MVPA and TV viewing. One might only speculate that children with high EE could compensate for this behaviour with a higher amount of MVPA. On the other hand, it may be that children with high MVPA eat more overall as well as during emotional situations. There is also a possibility that children with high MVPA have more hobbies, including those involving physical activity in competing environments, which might lead to more stress and subsequently EE. TV viewing is associated with higher intakes of sweet and salty snacks and carbonated beverages [55], and Ouwens et al. [54] found that the association between TV viewing and snacking was stronger in children with high EE. Distraction caused by the television can lead to diminished awareness of hunger and satiety, and eating in front of television can be considered “mindless eating”, which has been shown to be closely related to EE in children [56]. It is also possible that there is a third unmeasured variable explaining the association. For example, children with high negative affectivity could watch more TV as well as display more EE. 

We did not find an association between EE and BMI *z*-scores in the fully adjusted model. In line with our findings, other studies have also reported no difference in self-reported [20] and parent-reported [19] EE between normal weight and overweight 7 to 15-year-old children. However, several studies have also found a positive association [12,13,14,15,16,17,18]. In children, EE has been measured using self-reported or parent-reported questionnaires, and it has been shown that there is good concordance between these two approaches [13]. However, it seems that the majority of the studies reporting positive associations with BMI/being overweight have used parent-reported EE [12,14,15,16,17,18], and some studies using self-reported EE have even found an inverse association with BMI/being overweight [21,22,23]. It remains an interesting question for future studies as to whether inconsistent findings could be partially explained by the different reporting methods for EE.

We found a weak but significant negative association between self-reported EE and BMI *z*-scores in the unadjusted model, but after adjusting for other health behaviours (including MVPA), it was no longer significant. This result is probably due to including MVPA in the model, since EE was positively associated with MVPA in our sample, and MVPA was the strongest significant predictor of a lower BMI [36]. In earlier studies that reported a negative association [21,22,23], physical activity was not taken into account. Altogether, our results strengthen the existing literature that, in children, EE does not seem to be as clearly associated with obesity as in adults. 

It is interesting and somewhat controversial that EE was associated with the unhealthy diet pattern but not with BMI. However, in the ISCOLE sample, the unhealthy diet pattern was not associated with BMI [36]. It should be noted that the diet pattern scores describe dietary quality instead of energy intake per se, which may be more closely related to BMI. Nevertheless, it has been shown that food choices determined by dietary patterns track into adulthood [57]. It might be that EE already leads to the unhealthy diet pattern during childhood, but its consequences, such as excess weight gain, are visible only later in life. To our knowledge, there are no prospective studies examining these associations from childhood to adolescence or adulthood, but in one previous study, parent-reported EE of 5 to 6-year-old children predicted a higher BMI in 7 to 8-year-old children [25]. In adults, EE has also been shown to predict weight gain [10,11].

We did not find any interactions between EE and gender in association with dietary patterns, TV viewing, sleep duration, or BMI *z*-scores. Yet, the positive association between EE and MVPA was significantly stronger in boys. Further studies are needed to clarify and explain this observed interaction. In our study, there was no difference in reported EE between boys and girls, which is consistent with some previous findings [12,23], but not all, since Snoek et al. [22] found that 11–16-year-old girls reported more EE than boys. In site-specific analyses, we found that in the United Kingdom, higher amounts of EE were reported among girls than boys. The difference between mean EE scores was small (0.10 (0.55) in girls vs. −0.04 (0.50) in boys), and it is possible that the significant difference emerged by chance due to multiple testing (type I error).

A potential limitation of the present study was that EE was self-reported. However, Braet et al. [13] compared children’s self-reported EE vs. parent-reported EE and found that the agreement between reporting was good enough to conclude that it is possible to rely on either of the informants, especially for children aged 10 years and older. We measured EE using the EIES, which was developed in a similar age group as our sample. Even though the internal consistency was less than optimal for the four EIES items (Cronbach’s alpha value 0.6), the excellent fit of the one-factor CFA model and high factor loadings supported the unidimensionality of the scale. In addition, the percentage of children reporting no EE in our study was 32%. Similar percentages have been reported previously, regardless of the questionnaire or the reporter. In one study using the child-version of the Dutch Eating Behaviour Questionnaire to measure self-reported EE, 45% of the children reported that they never expressed EE [54], and in the other study, approximately 35% of the parents reported that their child never displays EE using the Children’s Eating Behavior Questionnaire [18]. As well as EE, diet was also self-reported. It has been suggested that in children aged 6 to 18 years, self-reporting of dietary intake is more valid than parental reporting [49,50]. Furthermore, in that age group, a relatively short (20–60 items) FFQ without requirement for portion size estimation, as used in the present study, has been suggested to be a valid method for measuring self-reported diet [49]. 

A further limitation was that the excluded children reported more EE, had higher scores on the unhealthy dietary pattern, watched more TV, and had higher BMI *z*-scores. This finding fits well with the hypothesis that unhealthy behaviours tend to cluster [33,34]. It is possible that this selection effect has attenuated the observed associations, and it may affect the generalizability of the results, although the observed differences between two groups were quite small, albeit statistically significant. Furthermore, it should also be mentioned that children’s opportunities to eat in response to negative emotions depend on how often they experience these emotions. Unfortunately, no measures of emotions, mood, or stress were included in the present study, because it was a secondary analysis of existing ISCOLE, and the questionnaire was originally designed to address the main research questions. Finally, the cross-sectional nature of the data does not allow any conclusions about the causality of the observed associations or their direction to be formed. However, a particular strength and novel aspect of this study is that the same EE questionnaire and outcome measures were used in a large and truly international study sample. We were also able to study multiple health behaviours simultaneously, and MVPA, sleep duration, and BMI were measured objectively.

## 5. Conclusions

In conclusion, we found a significant positive association between EE and an unhealthy diet pattern, which was consistent across the 12 different study sites. As previous studies have been almost completely focused on Western countries, this extends the present knowledge by suggesting that the association between EE and unhealthy diet pattern is not restricted to Western countries and their cultural and food environments. It is possible that EE can already lead to an unhealthy diet pattern during childhood, but its consequences, such as excess weight gain, are visible only later in life. The associations between EE and other health behaviours were either inconsistent between sites or not significant. We observed no association between EE and BMI *z*-scores. Prospective studies in different cultural contexts are needed to determine whether higher EE in children leads to an undesirable diet and subsequent obesity over time.

## Figures and Tables

**Table 1 nutrients-11-00351-t001:** Confirmatory factor analysis ^a^ for Emotion-Induced Eating Scale (EIES) negative emotion items.

	*n*	χ^2^ (df)	*p*-Value	CFI	TLI	RMSEA	SRMR	Standardized Factor Loadings for EIES Items ^b^			
								Sad	Worried	Mad	Bored
All	7319	2.66 (2)	0.264	1.00	1.00	0.01	0.01	0.79	0.69	0.62	0.46
**Gender**											
Boys	3393	0.32 (2)	0.854	1.00	1.00	<0.01	<0.01	0.79	0.70	0.61	0.44
Girls	3926	6.53 (2)	0.038	1.00	0.99	0.02	0.013	0.78	0.69	0.63	0.49
**Country (city/cities)**											
Australia (Adelaide)	526	0.89 (2)	0.642	1.00	1.01	<0.01	0.01	0.82	0.69	0.65	0.52
Brazil (Sao Paulo)	569	0.26 (2)	0.878	1.00	1.01	<0.01	0.01	0.75	0.70	0.43	0.85
Canada (Ottawa)	566	0.14 (2)	0.932	1.00	1.02	<0.01	0.01	0.94	0.66	0.66	0.41
China (Tianjin)	549	3.27 (2)	0.195	1.00	0.99	0.03	0.03	0.75	0.61	0.83	0.37
Colombia (Bogota)	919	4.20 (2)	0.123	0.99	0.96	0.04	0.03	0.61	0.55	0.53	0.36
Finland (Helsinki, Espoo, Vantaa)	535	1.29 (2)	0.524	1.00	1.01	<0.01	0.02	0.72	0.79	0.58	0.59
India (Bangalore)	620	0.79 (2)	0.675	1.00	1.03	<0.01	0.02	0.62	0.60	0.48	0.44
Kenya (Nairobi)	559	5.11 (2)	0.078	0.99	0.95	0.05	0.03	0.85	0.58	0.47	0.31
Portugal (Porto)	777	3.42 (2)	0.181	1.00	1.00	0.03	0.02	0.74	0.81	0.79	0.87
South Africa (Cape Town)	541	7.31 (2)	0.026	0.98	0.95	0.07	0.04	0.81	0.62	0.62	0.38
United Kingdom (Bath and Somerset)	525	1.08 (2)	0.583	1.00	1.01	<0.01	0.01	0.85	0.69	0.63	0.52
Unites States of America (Baton Rouge)	633	2.02 (2)	0.365	1.00	1.00	<0.01	0.02	0.84	0.76	0.76	0.60

(a) The weighted least squares estimation with robust standard errors and a mean and variance adjusted test statics was used, and items were used as ordinal variables. The model fit was evaluated with several types of fit indices including Chi-Square statistics, the Comparative Fit Index (CFI), the Tucker–Lewis Index (TLI), the Root Mean Square Error of Approximation (RMSEA), the and Standardized Root Mean Square Residual (SRMR). As suggested by Hu and Bentler [37], CFI and TLI values ≥0.95, RMSEA values ≤0.06, and SRMR values ≤0.08 were considered to indicate a good fit for the data. Results are presented for pooled data and separately for each gender and country (city/cities). (b) Sad = “I eat more when I’m sad”, worried = “I eat more, when I’m worried”, mad = “I eat when I’m mad”, bored = “I eat more, when I’m bored”.

**Table 2 nutrients-11-00351-t002:** Descriptive results of the analytical sample by gender and country (city/cities).

	Number of Participants (% Girls)	Mean (SD)								Household Classified Lowest Income ^e^, *n* (%)	Household Classified Highest Income ^e^, *n* (%)
		Emotional Eating Score ^a^	Unhealthy Diet Score ^b^	Healthy Diet Score ^b^	MVPA (min/day) ^c^	Sleep (min/day) ^c^	TV Viewing ^d^	BMI *z*-Score	Age (years)		
All	5426 (55)	0.03 (0.56)	−0.07 (0.93)	0.00 (0.99)	60 (25)	528 (53)	1.7 (1.2)	0.43 (1.25)	10.4 (0.6)	1376 (25)	1466 (27)
**Gender**											
Boys	2461 (0)	0.03 (0.56)	0.00 (0.98)	−0.06 (1.00)	70 (26)	524 (52)	1.8 (1.2)	0.53 (1.30)	10.4 (0.6)	596 (24)	696 (28)
Girls	2965 (100)	0.03 (0.56)	−0.14 (0.89)	0.04 (0.99)	52 (21)	532 (53)	1.6 (1.2)	0.36 (1.21)	10.4 (0.6)	780 (26)	770 (26)
**Country (city/cities)**											
Australia (Adelaide)	407 (54)	0.05 (0.54)	−0.30 (0.73)	0.24 (0.94)	65 (23)	565 (43)	1.7 (1.1)	0.57 (1.13)	10.7 (0.4)	88 (22)	94 (23)
Brazil (Sao Paulo)	378 (50)	0.24 (0.61)	0.09 (0.90)	−0.45 (1.05)	60 (27)	512 (49)	2.2 (1.4)	0.92 (1.43)	10.5 (0.5)	144 (38)	52 (14)
Canada (Ottawa)	481 (59)	−0.10 (0.51)	−0.50 (0.57)	0.49 (0.98)	59 (20)	544 (51)	1.4 (1.2)	0.42 (1.21)	10.5 (0.4)	88 (18)	186 (39)
China (Tianjin)	430 (48)	−0.09 (0.47)	−0.24 (0.96)	0.05 (0.90)	45 (16)	527 (39)	1.2 (1.1)	0.73 (1.54)	9.9 (0.5)	86 (20)	135 (31)
Colombia (Bogota)	810 (51)	−0.02 (0.52)	−0.08 (0.55)	−0.45 (0.74)	68 (25)	525 (49)	2.1 (1.1)	0.20 (1.05)	10.5 (0.6)	279 (34)	185 (23)
Finland (Helsinki, Espoo, Vantaa)	426 (55)	−0.08 (0.52)	−0.57 (0.43)	−0.15 (0.85)	70 (27)	508 (56)	1.4 (0.9)	0.27 (1.04)	10.5 (0.4)	81 (19)	174 (41)
India (Bangalore)	500 (55)	0.06 (0.51)	−0.10 (0.83)	−0.09 (0.89)	48 (21)	516 (44)	1.2 (0.9)	0.22 (1.36)	10.4 (0.5)	121 (24)	188 (38)
Kenya (Nairobi)	434 (54)	0.19 (0.57)	0.11 (1.01)	0.28 (0.99)	73 (32)	515 (52)	1.6 (1.3)	−0.06 (1.20)	10.2 (0.7)	101 (23)	126 (29)
Portugal (Porto)	490 (57)	−0.09 (0.56)	−0.36 (0.63)	0.25 (1.04)	56 (22)	497 (51)	1.5 (1.0)	0.83 (1.13)	10.4 (0.3)	96 (20)	105 (21)
South Africa (Cape Town)	267 (60)	0.35 (0.67)	1.08 (1.25)	0.26 (1.08)	62 (25)	552 (43)	2.0 (1.3)	0.20 (1.27)	10.2 (0.7)	133 (45)	37 (13)
United Kingdom (Bath and Somerset)	355 (57)	0.04 (0.53)	−0.17 (0.73)	0.03 (0.91)	64 (23)	569 (43)	1.7 (1.0)	0.40 (1.08)	10.9 (0.4)	95 (27)	81 (23)
Unites States of America (Baton Rouge)	418 (60)	0.00 (0.59)	0.59 (1.36)	−0.14 (1.14)	50 (19)	533 (55)	2.0 (1.4)	0.71 (1.27)	9.9 (0.6)	64 (15)	103 (25)

(a) Confirmatory factor analysis with weighted least squares estimation with robust standard errors and a mean and variance adjusted test statistic was conducted. Factor scores were computed using the Empirical Bayes approach, and the range of scores was −0.55 to 1.93. (b) Component scores identified with principal components analysis with an orthogonal varimax rotation. (c) The average amount of moderate to vigorous physical activity (MVPA) during the day and the average amount of night-time sleep, measured with accelerometers. (d) TV viewing score obtained from a questionnaire: minimum 0 points, maximum 5 points. (e) Annual household income was reported by parents using site-specific categories, which were merged into four levels.

**Table 3 nutrients-11-00351-t003:** The associations ^a^ between emotional eating factor scores (as the independent variable) and outcome variables ^b^. *n* = 5426.

	Unhealthy Diet Pattern		Healthy Diet Pattern		MVPA		Sleep		TV Viewing		BMI *z*-Score	
	Beta (SE)	*p*-Value	Beta (SE)	*p*-Value	Beta (SE)	*p*-Value	Beta (SE)	*p*-Value	Beta (SE)	*p*-Value	Beta (SE)	*p*-Value
Unadjusted ^c^	0.33 (0.02)	<0.0001	−0.03 (0.02)	0.739	2.95 (0.55)	<0.0001	−0.04 (1.20)	0.977	0.16 (0.03)	<0.0001	−0.09 (0.03)	0.004
1. Adjusted model ^d^	0.32 (0.02)	<0.0001	−0.03 (0.02)	0.291	2.65 (0.51)	<0.0001	0.01 (1.19)	0.993	0.15 (0.03)	<0.0001	−0.09 (0.03)	0.004
2. Adjusted model ^e^	0.29 (0.02)	<0.0001	−0.04 (0.02)	0.138	2.01 (0.51)	<0.0001	0.65 (1.22)	0.594	0.07 (0.03)	0.010	−0.06 (0.03)	0.055

(a) Analyzed with a multilevel (site, schools nested within sites) linear regression. (b) The unhealthy and healthy diet pattern scores identified using principal components analysis, daily moderate to vigorous physical activity (min, MVPA), nightly sleep duration (min), TV viewing scores, and BMI *z*-scores. (c) Only using the emotional eating factor score as an independent variable. (d) Adjusted by age, gender, and household income. (e) Adjusted by age, gender, household income, and other outcome variables.

**Table 4 nutrients-11-00351-t004:** The association between emotional eating ^a^ and outcome variables ^b^ in gender-stratified ^c^ and site-stratified ^d^ analyses.

	*N*	Unhealthy Diet Pattern		Healthy Diet Pattern		MVPA		Sleep		TV Viewing		BMI *z*-Score	
		Beta (SE)	*p*-Value	Beta (SE)	*p*-Value	Beta (SE)	*p*-Value	Beta (SE)	*p*-Value	Beta (SE)	*p*-Value	Beta (SE)	*p*-Value
**Gender**													
Boys	2461	0.29 (0.03)	<0.0001	−0.03 (0.04)	0.346	3.31 (0.86)	0.0001	−0.91 (1.80)	0.613	0.08 (0.04)	0.068	−0.11 (0.05)	0.014
Girls	2965	0.30 (0.03)	<0.0001	−0.03 (0.03)	0.392	1.28 (0.60)	0.035	1.60 (1.67)	0.337	0.05 (0.04)	0.157	−0.03 (0.04)	0.493
**Study sites**													
Australia (Adelaide)	407	0.35 (0.06)	<0.0001	−0.01 (0.09)	0.911	0.00 (1.95)	0.998	−2.43 (4.01)	0.544	0.09 (0.10)	0.390	0.17 (0.11)	0.117
Brazil (Sao Paulo)	378	0.19 (0.08)	0.014	−0.06 (0.09)	0.546	3.60 (1.92)	0.062	3.95 (4.04)	0.329	0.05 (0.12)	0.674	0.07 (0.12)	0.562
Canada (Ottawa)	481	0.18 (0.05)	0.0001	−0.01 (0.09)	0.903	1.06 (1.63)	0.515	−0.69 (4.08)	0.867	0.27 (0.10)	0.008	−0.09 (0.11)	0.392
China (Tianjin)	378	0.43 (0.10)	<0.0001	−0.04 (0.10)	0.710	−0.75 (1.57)	0.634	0.66 (4.17)	0.874	0.34 (0.10)	0.001	−0.19 (0.16)	0.227
Colombia (Bogota)	810	0.20 (0.04)	<0.0001	0.10 (0.05)	0.045	3.33 (1.51)	0.028	2.97 (3.44)	0.389	0.10 (0.08)	0.186	−0.16 (0.07)	0.021
Finland (Helsinki, Espoo, Vantaa)	426	0.18 (0.04)	<0.0001	0.02 (0.08)	0.809	1.09 (2.25)	0.630	−2.47 (5.15)	0.632	−0.04 (0.09)	0.624	−0.21 (0.10)	0.032
India (Bangalore)	500	0.21 (0.07)	0.004	−0.10 (0.08)	0.197	−0.10 (1.50)	0.945	1.24 (3.98)	0.756	−0.04 (0.08)	0.583	−0.14 (0.12)	0.246
Kenya (Nairobi)	434	0.30 (0.08)	0.0003	−0.09 (0.09)	0.283	2.80 (1.89)	0.139	1.08 (4.71)	0.819	0.01 (0.11)	0.933	0.12 (0.10)	0.243
Portugal (Porto)	490	0.27 (0.05)	<0.0001	0.05 (0.09)	0.587	2.86 (1.56)	0.068	−0.51 (4.37)	0.907	0.13 (0.08)	0.110	−0.11 (0.10)	0.287
South Africa (Cape Town)	267	0.62 (0.11)	<0.0001	−0.21 (0.11)	0.052	5.12 (2.23)	0.022	2.41 (4.46)	0.589	0.09 (0.13)	0.475	0.06 (0.13)	0.633
United Kingdom (Bath and Somerset)	355	0.23 (0.07)	0.0007	0.01 (0.09)	0.913	−1.98 (1.97)	0.317	−4.68 (4.27)	0.273	−0.04 (0.10)	0.684	0.01 (0.11)	0.920
Unites States of America (Baton Rouge)	418	0.47 (0.09)	<0.0001	−0.03 (0.10)	0.788	0.53 (1.44)	0.715	1.78 (4.52)	0.695	−0.03 (0.11)	0.791	−0.06 (0.11)	0.600

(a) Emotional eating factor scores as the independent variable. (b) The unhealthy and healthy diet pattern scores identified using principal components analysis, daily moderate to vigorous physical activity (min, MVPA), nightly sleep duration (min), TV viewing scores, and BMI *z*-scores. (c) Analyzed with a multilevel (site, schools nested within sites) linear regression, adjusted with age, household income, and other outcome variables. (d) Analyzed with a multilevel (schools) linear regression, adjusted by age, gender, household income, and other outcome variables.

**Table 5 nutrients-11-00351-t005:** The associations ^a^ between categorized emotional eating factor scores ^b^ and outcome variables ^c^.

	*n* (%)	Unhealthy Diet Pattern		Healthy Diet Pattern		MVPA		Sleep		TV Viewing		BMI *z*-Score	
		Beta (SE)	*p*-Value	Beta (SE)	*p*-Value	Beta (SE)	*p*-Value	Beta (SE)	*p*-Value	Beta (SE)	*p*-Value	Beta (SE)	*p*-Value
**All**			<0.0001		0.190		0.0008		0.105		0.002		0.248
No emotional eating	1759 (32)	ref.		ref.		ref.		ref.		ref.		ref.	
1. quarter	943 (17)	0.11 (0.03)	0.0006	−0.05 (0.04)	0.201	−0.15 (0.82)	0.853	4.42 (1.95)	0.023	0.11 (0.05)	0.014	−0.09 (0.05)	0.065
2. quarter	771 (14)	0.14 (0.03)	<0.0001	−0.08 (0.04)	0.044	0.51 (0.86)	0.551	−1.20 (2.05)	0.557	0.08 (0.05)	0.105	−0.01 (0.05)	0.889
3. quarter	1017 (19)	0.27 (0.03)	<0.0001	−0.06 (0.04)	0.145	1.27 (0.79)	0.109	2.28 (1.89)	0.230	0.18 (0.04)	<0.0001	−0.06 (0.05)	0.239
4. quarter	936 (17)	0.43 (0.03)	<0.0001	−0.08 (0.04)	0.050	3.36 (0.84)	<0.0001	1.74 (2.01)	0.387	0.09 (0.05)	0.047	−0.09 (0.05)	0.093
**Boys**			<0.0001		0.573		0.001		0.833		0.097		0.052
No emotional eating	789 (32)	ref.		ref.		ref.		ref.		ref.		ref.	
1. quarter	432 (18)	0.10 (0.05)	0.060	−0.03 (0.06)	0.656	−0.96 (1.37)	0.486	−0.09 (2.87)	0.974	0.06 (0.07)	0.371	−0.17 (0.07)	0.021
2. quarter	356 (15)	0.15 (0.05)	0.007	−0.08 (0.06)	0.190	1.45 (1.45)	0.316	−2.40 (3.03)	0.427	0.05 (0.07)	0.512	−0.10 (0.08)	0.215
3. quarter	452 (18)	0.29 (0.05)	<0.0001	−0.08 (0.06)	0.182	2.28 (1.35)	0.093	0.74 (2.83)	0.795	0.19 (0.07)	0.006	−0.17 (0.07)	0.020
4. quarter	432 (18)	0.41 (0.05)	<0.0001	−0.07 (0.06)	0.261	5.09 (1.42)	0.0003	−2.14 (2.96)	0.470	0.09 (0.07)	0.224	−0.18 (0.08)	0.020
**Girls**			<0.0001		0.615		0.046		0.025		0.038		0.737
No emotional eating	970 (33)	ref.		ref.		ref.		ref.		ref.		ref.	
1. quarter	511 (17)	0.12 (0.04)	0.004	−0.06 (0.05)	0.265	−0.08 (0.96)	0.932	8.01 (2.67)	0.003	0.14 (0.06)	0.017	−0.02 (0.07)	0.727
2. quarter	415 (14)	0.15 (0.04)	0.001	−0.06 (0.06)	0.256	−0.34 (1.01)	0.736	−0.46 (2.81)	0.870	0.09 (0.06)	0.149	0.07 (0.07)	0.321
3. quarter	565 (19)	0.25 (0.04)	<0.0001	−0.03 (0.05)	0.610	−0.08 (0.93)	0.932	3.48 (2.58)	0.177	0.16 (0.06)	0.005	0.02 (0.06)	0.727
4. quarter	504 (17)	0.45 (0.04)	<0.0001	−0.07 (0.05)	0.193	2.62 (1.00)	0.009	4.24 (2.77)	0.126	0.07 (0.06)	0.240	−0.03 (0.07)	0.700

(a) Analyzed with multilevel (site, schools nested within sites) linear regression. All models adjusted with age, household income, other outcome variables, and model for all participants adjusted also for gender. (b) The emotional eating factor scores were categorized by first setting the participants with no emotional eating as reference group and then dividing rest of the participants to quarters based on emotional eating factor scores. The emotional eating score in “No emotional eating” -group was −0.552 and minimum and maximum values of emotional eating factor score was −0.26 and −0.06 in 1st quarter, −0.05 and 0.20 and in 2nd quarter, 0.21 and 0.65 in 3rd quarter, and 0.66 and 1.93 in 4th quarter. (c) The unhealthy and the healthy diet pattern scores identified using principal components analysis, daily moderate to vigorous physical activity (min, MVPA), nightly sleep duration (min), TV viewing scores, and BMI *z*-scores.

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
