# Peer review of "Emotional Eating, Health Behaviours, and Obesity in Children: A 12-Country Cross-Sectional Study"

_nutrients, 2019, doi:10.3390/nu11020351_

Round 1

Reviewer 1 Report

Jalo and colleagues report the results of a study testing for relationships between body weight, dietary habits, eating in response to negative emotions and demographic factors in 9-11 year-old children living in 12 countries. Consistent associations are found between EE and uptake of an unhealthy diet, but not with uptake of a healthy diet, physical activity, sleep, TV viewing behaviour or BMI z-score. The manuscript is clearly written. One major query (detailed below) concerns the BMI z-score data reported in Table 3.

Major comments

1)      The most consistent association is between EE and unhealthy diet patterns. Might this not simply reflect how ‘discretionary’ type foods are consumed? That is, in opportunistic settings when one looks for food that is quick to obtain. It is interesting that EE was not associated with uptake of a healthy diet.

2)      Table 2: could the authors please explain the emotional eating score in greater detail – how is this calculated from the CFA and what is the range of values it can take? Data also indicate variability in predictor and outcome variables between countries – were these statistically significant?

3)      A noteworthy aspect of the results is that the relationship between EE and unhealthy diet intake was positive across all sites. Does the inclusion of ‘study site’ as the highest cluster in the regression models allow for analysis of whether differences between sites were statistically significant?

4)      Line 254: given the ‘skewness of the original continuous variable’, would it not be more appropriate to include the categorical analysis in the main text and report the analysis using the continuous measure in the Supplementary materials?  This would seem the case given that 57% of the sample reported very low levels of emotional eating (32% ‘never or almost never’ for all Qs, 25% ‘never or almost never’ for ¾ Qs).

5)      Table 3: the data for BMI z-score require clarification. Is it true that the beta (-0.09), s.e (0.03) and p-value (0.004) are identical for the unadjusted and first adjusted model? Additionally, why were BMI z-scores analysed using adjusted model 2 if this model controls for BMI z-scores? Adjusted model 1 seems more appropriate here and, if true, shows a significant inverse association with EE (a very interesting result that would warrant discussion).

Minor comments

Abstract: here the effects of gender (generally null) should be reported.

Lines 276 and 285-288: as above, is the fully-adjusted model appropriate for BMI z-scores?

Lines 288-292: what might explain the positive association between EE and MVPA?

It is somewhat surprising that no measures of mood, emotionality or stress were taken, since the opportunity to eat when mad/sad/worried/bored critically depends on how often children experienced these emotions.

Lines 298-299: please provide the age ranges for these studies.

Line 299: typo – ‘On’ should be ‘One’.

Line 311: the apparent negative relationship between EE and BMI z-scores is worth discussing here, together with the MVPA data, which track positively with EE (and presumably negatively with BMI).

Lines 319-320 (and lines 55-57 of introduction): one wonders about the validity of measuring EE in 5-6 year-old children. First, is there evidence that these constructs can be grasped by children of this age, let alone measured accurately? Even if so, do 5-6 year-old children have opportunities to procure food when they are in negative emotional states?

Lines 360-361: please clarify what is meant by ‘emerged by chance due to multiple testing’.

Lines 394-395: is it not possible that an unhealthy diet promotes EE as well as EE promoting an unhealthy diet?

Author Response

Response to Reviewer 1 comments

Major comments

Point 1: The most consistent association is between EE and unhealthy diet patterns. Might this not simply reflect how ‘discretionary’ type foods are consumed? That is, in opportunistic settings when one looks for food that is quick to obtain. It is interesting that EE was not associated with uptake of a healthy diet.

Response 1: Thank you for this very good suggestion. It may indeed be one possible explanation. As such, we have added the following: “It must be noted that in addition to the palatability and nutrient composition of foods, food choice in response to emotional states might also reflect the availability and accessibility of the foods. Many foods included in the unhealthy diet pattern (i.e. fast foods, ice cream, fried food, French fries, potato chips, cakes and sugar-sweetened sodas) are ready to eat and easily obtainable even for 9–11-year-olds.” to the Discussion section (lines 327–331).

Point 2:   Table 2: could the authors please explain the emotional eating score in greater detail – how is this calculated from the CFA and what is the range of values it can take? Data also indicate variability in predictor and outcome variables between countries – were these statistically significant?

Response 2: Thank you for pointing this out. Factor scores were computed from the CFA using Empirical Bayes Modal approach. We have added the information to the Methods section (lines 158–159), as well as to the Table 2. In addition, we included the range of factor scores (-0,52–1,93) to the Table 2. The Reviewer is correct, there is variability between countries in emotional eating scores as well as outcome variables. The statistical significance of these differences was not tested, since the main research question of this paper was to study the associations between variables.

Point 3: A noteworthy aspect of the results is that the relationship between EE and unhealthy diet intake was positive across all sites. Does the inclusion of ‘study site’ as the highest cluster in the regression models allow for analysis of whether differences between sites were statistically significant?

Response 3: This is an interesting point. We did not test whether the differences in beta estimates between countries were statistically significant, because the large number of countries would mean so many comparisons that the results would have been difficult to handle. Furthermore, comparing each country against all of the remaining countries was outside of the scope of this paper. We have now included this explanation in the Discussion section (lines 347–350).

Point 4:      Line 254: given the ‘skewness of the original continuous variable’, would it not be more appropriate to include the categorical analysis in the main text and report the analysis using the continuous measure in the Supplementary materials?  This would seem the case given that 57% of the sample reported very low levels of emotional eating (32% ‘never or almost never’ for all Qs, 25% ‘never or almost never’ for ¾ Qs).

Response 4: We agree that the distribution of the emotional eating scores were skewed Yet, as there were no marked differences between the results from continuous and categorical analysis, we decided to use the continuous analysis to keep the Results table more comprehensible. However, and based on your comment, we have moved the results from the categorical analysis to the main paper (i.e., as Table 5).

Point 5: Table 3: the data for BMI z-score require clarification. Is it true that the beta (-0.09), s.e (0.03) and p-value (0.004) are identical for the unadjusted and first adjusted model? Additionally, why were BMI z-scores analysed using adjusted model 2 if this model controls for BMI z-scores? Adjusted model 1 seems more appropriate here and, if true, shows a significant inverse association with EE (a very interesting result that would warrant discussion).

Response 5: Regarding the results from unadjusted and first adjusted models, the results are identical when these values are rounded to two decimals. More precise values for the unadjusted are beta (s.e.) = -0.08898 (0.03054); p=0.0036 and for the first adjusted models beta (s.e.) = -0.08782 (0.03056); p=0.0041. Regarding the second point, we agree that it was not clearly written in the table which variables were included as covariates in each model.  In the case of BMI z-scores, the 2nd adjusted model controls for other outcome variables (i.e., dietary patterns, MVPA, sleep duration and TV-viewing). No additional adjustment for BMI z-scores was done. We have revised the Table 3 accordingly to make this clearer.

Minor comments

Point 6: Abstract: here the effects of gender (generally null) should be reported.

Response 6: We have added to the Abstract that the results tended to be similar in boys and girls (lines 36–37).

Point 7: Lines 276 and 285-288: as above, is the fully-adjusted model appropriate for BMI z-scores?

Response 7: As described in the point 5, the fully-adjusted model is appropriate for BMI z-score since it adjusts for other outcome variables and no additional adjustment for BMI z-scores is done.

Point 8: Lines 288-292: what might explain the positive association between EE and MVPA?

Response 8: We wanted to study different health behaviours in relation to emotional eating since unhealthy behaviours tend to cluster. The observed positive association between EE and MVPA was somewhat unexpected since it is against the hypothesis regarding clustering of unhealthy behaviours. We have speculated the potential explanations for this observed association in the Discussion section as follows: “One might only speculate that children with high EE could compensate for this behaviour with higher amount of MVPA. On the other hand, it may be that children with high MVPA eat more overall, also during emotional situations. There is also a possibility that children with high MVPA have more hobbies including physical activity in competing environments, which might lead to more stress and subsequently EE.” (lines 392–396). Yet, we tried to make it very clear that because our study is cross-sectional, we cannot make any conclusions about the direction of the association or possible underlying mechanisms (lines 387–389).

Point 9: It is somewhat surprising that no measures of mood, emotionality or stress were taken, since the opportunity to eat when mad/sad/worried/bored critically depends on how often children experienced these emotions.

Response 9: This is a very good point.The present study was a secondary analysis of this existing ISCOLE, which main aim of was to study different health behaviours in relation to obesity. Unfortunately, no measures of emotions, mood or stress were included in the study, because the questionnaire was originally designed for the main research questions. We have now acknowledged this in the Discussion section (lines 477–481).

Point 10: Lines 298-299: please provide the age ranges for these studies.

Response 10: Based on this very relevant suggestion, we have added the age range to the paper (line 336).

Point 11: Line 299: typo – ‘On’ should be ‘One’.

Response 11: Thank you for pointing this out, the typo has now been corrected (line 338).

Point 12: Line 311: the apparent negative relationship between EE and BMI z-scores is worth discussing here, together with the MVPA data, which track positively with EE (and presumably negatively with BMI).

Response 12: This refers to a sentence “It is interesting and somewhat controversial that EE was associated with the unhealthy diet pattern but not with BMI”. We modified the structure of the Discussion section to make it easier to follow. We now discuss the relationship between EE and BMI z-score together with MVPA data on lines 429-434, just before moving on to the point in question. MVPA indeed was negatively associated with BMI in ISCOLE sample (Katzmarzyk et al 2015). There was a negative association between EE and BMI z-score in 1st adjusted model, but after adjusting for other outcome variables (including the MVPA), the association was no longer significant.

Point 13: Lines 319-320 (and lines 55-57 of introduction): one wonders about the validity of measuring EE in 5-6 year-old children. First, is there evidence that these constructs can be grasped by children of this age, let alone measured accurately? Even if so, do 5-6 year-old children have opportunities to procure food when they are in negative emotional states?

Response 13: This is a good question. In these studies, the emotional over-eating in children was reported by the parents using the Child Eating Behaviour Questionnaire. The questionnaire has been developed by Wardle et al in 2001, and it has been widely used since then to measure different dimensions of eating style in young children. Despite the previous use of the questionnaire, it can be questioned how accurately parents can estimate the behaviour of their child. However, Braet et al (2007) found that there is moderate agreement between child’s and parent’s reports of emotional eating even in children as young as 7–9 years old.

Point 14: Lines 360-361: please clarify what is meant by ‘emerged by chance due to multiple testing’.

Response 14: We have now clarified this in the manuscript (line 455). When multiple test are performed, it is possible that some of these give a significant p-values just by chance even though the null hypothesis was true. So, there is a chance for type I error (“false positive”).

Point 15: Lines 394-395: is it not possible that an unhealthy diet promotes EE as well as EE promoting an unhealthy diet?

Response 15: This is a good additional perspective and we admit overlooking it in our manuscript. We have now added following to the Discussion section (lines 331–335): “In addition, because our study, as well as most of the earlier research regarding the association between emotional eating and diet, was cross-sectional, we cannot rule out the possibility of reverse relationship. That might be the case if unhealthier diet is associated with decreased mood, or decreased hunger control, making it easier to eat in emotional states.”

Reviewer 2 Report

The author investigated association between emotional eating and obesity in children.  Surveys were conducted in 12 countries from various region of the world and there observed no association between emotional eating and obesity.  

Line 29: There is no need to describe the definition of BMI.  

Table 2: Please describe that country-specific income categories were merged into four levels.  

Line 287-291, 321-339 and Table 3: Please add convincing explanations about the reason why there is an association between emotional eating factor score and daily moderate to vigorous physical activity.  

Line 340-352: Please give enough explanation that both MVPA and TV viewing positively associated with emotional eating.  

Line 362-377: Please describe the reliability of self reported emotional eating score for 9 to 11-year-old children though cited literatures were studies in 7-15 years and 6-18 years.  

Author Response

Response to Reviewer 2 comments

Point 1: Line 29: There is no need to describe the definition of BMI.  

Response 1: We agree, and we have removed the definition from the manuscript.

Point 2: Table 2: Please describe that country-specific income categories were merged into four levels.  

Response 2: Thank you, this is a good suggestion. We have added this information to the footer of Table 2.

Point 3: Line 287-291, 321-339 and Table 3: Please add convincing explanations about the reason why there is an association between emotional eating factor score and daily moderate to vigorous physical activity.  

Response 3: We wanted to study different health behaviours in relation to emotional eating since they tend to cluster. The observed positive association between EE and MVPA was somewhat unexpected since it is against the hypothesis regarding clustering of unhealthy behaviours. We have speculated the potential explanations for this observed association in the Discussion section, Discussion section as follows: “One might only speculate that children with high EE could compensate for this behaviour with higher amount of MVPA. On the other hand, it may be that children with high MVPA eat more overall, also during emotional situations. There is also a possibility that children with high MVPA have more hobbies including physical activity in competing environments, which might lead to more stress and subsequently EE.” (lines 392–396). Yet, we tried to make it very clear that because our study is cross-sectional, we cannot make any conclusions about the direction of the association or possible underlying mechanisms (lines 390–392).

Point 4: Line 340-352: Please give enough explanation that both MVPA and TV viewing positively associated with emotional eating.  

Response 4: Analyses regarding both MVPA and TV viewing were exploratory and based on the hypothesis of the clustering of unhealthy behaviours. Because of limited number of previous studies and cross-sectional nature of our data, we cannot provide mechanistic explanations why these variables are positively associated with emotional eating. Yet, we have speculated the possible mechanisms in the Discussion section (lines 392–414). In addition, we have added to the manuscript (lines 390–392) that determining the causal relationship and possible mechanisms for the association between emotional eating and both MVPA and TV viewing remains a question for future studies.

Point 5: Line 362-377: Please describe the reliability of self reported emotional eating score for 9 to 11-year-old children though cited literatures were studies in 7-15 years and 6-18 years.  

Response 5: We agree that it can be questioned, how well children can reflect upon their emotions, as well as the association between emotions and eating. We have mentioned in the manuscript (lines 456–459) that EE has been measured using self-reported or parent-reported questionnaires in children, and it has been shown that there is good agreement between these two approaches, especially for children aged 10 years and older (Braet et al 2007).

Reviewer 3 Report

I wrote my comments and suggestions in the attached document.

Author Response

Response to Reviewer 3 Comments

Point 1: - the way emotional eating is measured. I do not completely understand the reasons for excluding three of the seven original items from the questionnaire other than the authors not agreeing with the scale. Previous work of Striegel et al confirmed the validity of this scale for the population under investigation. Even more so, the reliability of the four item version is not acceptable (.60), while the original version was acceptable. The authors should include the remaining three items and only exclude these if (factor and reliability) analyses suggest these to be redundant or to measure a different construct. In addition, I find the low reliability to easily dismissed in the discussion part because of an excellent fit with the one-factor model.

Response 1: Thank you for raising this important point. We apologize for not being clear why we decided to reduce the number of items. The main reason was conceptual. “Emotional eating” is traditionally defined as eating in response to negative emotions. We wanted to follow that definition in our paper, and we have now clarified it in the manuscript (lines 131–132). Even though we made the initial decision to exclude three items based on the conceptual reasons, we wanted to test also the original scale for unidimensionality. We performed the CFA for 7-item model. Model fit was better for the 4-item model based on  χ2-statistics. For the 7-item model χ2 (df) was 364.68 (14) (p<0.001) whereas for 4-item model respective values were 2.66 (2) (p=0.264). Moreover, in our sample, the reliability as measured with the Cronbach’s alpha was not much higher in 7-item scale compared to 4-item scale (0.65 vs. 0.60), even though in Striegel-Moore’s original work it was 0.78.

Point 2: - the theoretical contextualization in the introduction of the paper. The authors rightfully mention the inconsistencies in the literature regarding the association between emotional eating and obesity in children, but a factor they do not seem to consider are differences in type of measure used to assess emotional eating. So how does measurement variability fit the inconsistencies found in the literature?

Response 2: This is a relevant point, and we now consider differences in type of measure used to assess emotional eating (parent-reported vs. self-reported) in the Introduction section (lines 62–67).

Point 3: - A study not referred to in the manuscript is the study of Coumans et al 2018 in the International Journal of Obesity, although the study seems highly relevant considering that it is about emotion-driven impulsiveness (so a more general measure and specifically focused on eating behavior, although emotion induced eating may be such a behavior) and its relation with weight in a large European adolescent sample. In line with this, hardly any explanation is provided about how and why emotional eating may lead to unhealthy behaviors and/or higher weight. What are the psychological and physiological mechanisms underlying this association? This is important as it may also shed light on the inconsistencies found in the literature.

Reponse 3: Thank you for pointing this out. We included the study of Coumans et al 2018 to our references as well as elaborated the possible explanations why emotional eating would be associated with food choices and weight (lines 49–55 and 67–70 in the Introduction section). We feel that this greatly improved the Introduction.

Point 4: - an incorrect reference in the discussion. On page 11 line 277 the authors refer to the work of van Strien et al as evidence for a lack of association between emotional eating and weight. However, this article does report an association between the concept of emotional eating and weight. Furthermore, the second reference, Caccialanza et al who investigated parent report of emotional eating in children did only test differences in emotional eating between different weight groups (normal weight versus overweight) and did not examine the relation with weight as a continuous variable. The risks of using such a categorized variable include both information loss and biased information as all people in that weight group (for example someone with e BMI of 25.3 is consider similar to a person with BMI of 29.7) end up in the same weight group due to classification. This may result in a loss of power to detect an effect.

Response 4: Thank you for pointing this out. There was indeed an incorrect reference. Instead of number van Strien et al 2009 (originally number 7 and number 6 in the revised vesion) it should have been van Strien et al 2008 (number 19 in the revised version), which compared the emotional eating scores of normal weight and overweight children. Furthermore, both these studies showing no association do compare emotional eating score between different weight group which indeed may result in information loss. We have changed the text to more precise wording (lines 419–420).

Round 2

Reviewer 2 Report

The revised manuscript has been properly modified according to the referee's pointed out. 

Line 29:  I can see the definition of BMI in abstract section.  In the present review letter, I have proposed  to remove the definition of BMI in abstract.  

Author Response

Response to Reviewer 2 Comments

Point 1: Line 29:  I can see the definition of BMI in abstract section.  In the present review letter, I have proposed to remove the definition of BMI in abstract.

Response 1: We agree that we have made a mistake with this. We originally confused this with line 57, where the abbreviation of BMI is written out in full. We were going to remove that and only use the abbreviation, but unfortunately we forgot to do that. Regarding the Abstract section, there is written “BMI was calculated using measured weight and height.” We think that this does not refer to the definition of BMI, but that the weight and height were measured and not for example self-reported. Therefore, we would like to keep this sentence in the Abstract section, and not do any further changes to the manuscript.

Reviewer 3 Report

My concerns were nicely addressed by the authors and I think the manuscript was greatly improved.

I would like to recommend that the authors mention the CFA for the 7-item model also in the manuscript and not only in the "responses to the reviewer" letter, as I think it supports their choice to reduce the questionnaire from 7 to 4 items, in particular since the reliability of the questionnaire is rather weak.

I thank the authors for including the Coumans et al paper. However, I still think this paper can be better linked to the present findings and rational (it now seems the authors included the paper to satisfy me as a reviewer, that was not the purpose of me mentioning this paper. I really think the concepts are linked and the Coumans et al findings have value for the discussion of the results), but I leave this up to the authors. 

Author Response

Response to Reviewer 3 Comments

Point 1: I would like to recommend that the authors mention the CFA for the 7-item model also in the manuscript and not only in the "responses to the reviewer" letter, as I think it supports their choice to reduce the questionnaire from 7 to 4 items, in particular since the reliability of the questionnaire is rather weak.

Response 1: We agree that this is a relevant suggestion and we have completed the manuscript accordingly as follows (lines 145-150): “For comparison, we tested also the original 7-item scale. CFA indicated poorer fit compared to 4-item model [χ2(df)=364.68(14), p for χ2<0.001, CFI=0.96, TLI=0.94, RMSEA=0.06, and SRMR=0.04]. Furthermore, Cronbach’s alpha for the 7-item scale (0.65) was not substantially higher compared to the 4-item scale, especially because alpha increases always when more items are added [38]. These results further supported our initial decision to include only the four items measuring eating in response to negative emotions.”

Point 2: I thank the authors for including the Coumans et al paper. However, I still think this paper can be better linked to the present findings and rational (it now seems the authors included the paper to satisfy me as a reviewer, that was not the purpose of me mentioning this paper. I really think the concepts are linked and the Coumans et al findings have value for the discussion of the results), but I leave this up to the authors.

Response 2: Thank you for pointing this out. We agree that emotion-driven impulsiveness is likely to be linked with emotional eating, since difficulties in emotion regulation may be one possible mechanism underlying EE. We have now made this clearer in the Introduction (lines 50–52) to better link the results by Coumans et. al. to the rationale of our manuscript. We also found another paper by Coumans et al 2018 in Appetite, which reported that emotion-driven impulsiveness is associated with more frequent snacking, especially for energy-dense snacks. This paper is also relevant regarding our findings, and we have added that to our Discussion (lines 315-318). We think that these revisions further improved the rationale and the discussion of the results of the present study.